# Inhibition of Liquid–Liquid Phase Separation for Breaking the Solubility Barrier of Amorphous Solid Dispersions to Improve Oral Absorption of Naftopidil

**DOI:** 10.3390/pharmaceutics14122664

**Published:** 2022-11-30

**Authors:** Masafumi Fukiage, Kyosuke Suzuki, Maki Matsuda, Yohei Nishida, Michinori Oikawa, Takuya Fujita, Kohsaku Kawakami

**Affiliations:** 1Pharmaceutical R&D, Ono Pharmaceutical Co., Ltd., 3-3-1, Sakurai, Shimamoto-cho, Mishima-gun, Osaka 618-8585, Osaka, Japan; 2Pharmaceutical and ADMET Research Department, Daiichi Sankyo RD Novare Co., Ltd., 1-16-13, Kitakasai, Edogawa-ku, Tokyo 134-8630, Japan; 3Research & Development Division, Towa Pharmaceutical Co., Ltd., 134, Chudoji Minami-machi, Shimogyo-ku, Kyoto 600-8813, Kyoto, Japan; 4Technology Research & Development, Sumitomo Pharma Co., Ltd., 33-94, Enoki-cho, Suita, Osaka 564-0053, Osaka, Japan; 5Pharmaceutical Development Department, Sawai Pharmaceutical Co., Ltd., 5-2-30, Miyahara, Yodogawa-ku, Osaka 532-0003, Osaka, Japan; 6College of Pharmaceutical Sciences, Ritsumeikan University, 1-1-1 Noji-Higashi, Kusatsu, Kyoto 525-8577, Shiga, Japan; 7Research Center for Functionals Materials, National Institute for Materials Science, 1-1 Namiki, Tsukuba 305-0044, Ibaraki, Japan

**Keywords:** solid dispersion, amorphous, supersaturation, liquid–liquid phase separation, oral absorption, dissolution test

## Abstract

Amorphous solid dispersion (ASD) is one of the most promising technologies for improving the oral absorption of poorly soluble compounds. In this study, naftopidil (NFT) ASDs were prepared using vinylpyrrolidone-vinyl acetate copolymer (PVPVA), hydroxypropyl methylcellulose acetate succinate (HPMCAS), and poly(methacrylic acid-co-methyl methacrylate) L100-55 (Eudragit) to improve the dissolution and oral absorption behaviors of NFT. During the dissolution process of ASD, liquid–liquid phase separation (LLPS) may occur when certain requirements are met for providing a maximum quasi-stable concentration achievable by amorphization. The occurrence of LLPS was confirmed in the presence of PVPVA and HPMCAS; however, Eudragit inhibited LLPS owing to its molecular interaction with NFT. Although the dissolution behavior of the Eudragit ASD was found to be markedly poorer than that of other ASDs, it offered the best oral absorption in rats. The findings of the current study highlight the possibility for improving the oral absorption of poorly soluble drugs by this ASD, which should be eliminated from candidate formulations based on the conventional in vitro tests.

## 1. Introduction

Oral administration is the most common route of drug delivery, with drug absorption occurring via the gastrointestinal tract. The oral absorption of drugs is frequently limited by poor solubility and a poor dissolution rate. Accordingly, formulation technologies are required to assist in the dissolution process, which includes the use of salt, cocrystal, and enabling formulations, such as amorphous solid dispersion (ASD) and self-emulsifying dosage forms [1,2,3,4]. Establishing a supersaturated state in the gastrointestinal tract is recognized as advantageous [5,6] relative to the increase in equilibrium solubility using additives, where the interplay of solubility and permeability is anticipated [7]. ASD is a very effective formulation for improving oral absorption, where polymers play an important role in maintaining a supersaturated state in the gastrointestinal tract.

ASD can offer a supersaturated state during its dissolution if the dissolution rate is sufficiently fast and drug crystallization is not prompt. The highly supersaturated solution may be separated into concentrated and diluted phases to minimize the Gibbs energy [5,8]. Drug-rich droplets formed by liquid–liquid phase separation (LLPS) can act as a reservoir for maintaining the amorphous solubility [9,10,11] and/or can be a carrier that effectively diffuses in an unstirred water layer on the intestinal membrane [12,13]. In our previous studies [14,15,16], the oral absorption of poorly soluble drugs from ASDs exhibited a correlation with the LLPS concentrations, which might be a natural consequence if the absorption is controlled by drug solubility. The apparent LLPS concentration can be significantly influenced by coexisting molecules. Surfactants are examples that exhibit remarkable effects on apparent LLPS [9,14,15]. However, its effect cannot be understood in a simple manner. Bile salt is the most important class of surfactants, of which the role must be assessed to interpret the oral absorption of supersaturating dosage forms. Lu et al. evaluated the effect of six types of bile salts on LLPS and the membrane permeation behaviors of supersaturated telaprevir [17]. Based on their results, trihydroxy bile salts, including sodium taurocholate and sodium glycocholate, did not influence LLPS or transmembrane flux, regardless of their state (i.e., as a monomer or micelles), whereas dihydroxy bile salts, such as sodium taurodeoxycholate and sodium glycodeoxycholate increased the LLPS concentration and decreased transmembrane flux, respectively, only when they existed as micelles. Sodium taurochenodeoxycholate and sodium glycochenodeoxycholate may decrease LLPS; however, whether LLPS occurred was unclear. Nonetheless, regardless of the state of the bile salts, the transmembrane flux was found to decrease in the presence of these bile salts. Complexation of drugs with polymers also influences the transmembrane flux. According to Mosquera-Giraldo et al., the transmembrane flux of telaprevir was suppressed by molecular complexation with the relatively hydrophobic polymer, the cellulose derivative cellulose acetate suberate [18].

The formation of the LLPS state is important for achieving a stable supersaturated state. However, the LLPS concentration is the upper limit of the amount of molecularly dissolved drug, which suggests that oral absorption may be better if LLPS is inhibited. We employed naftopidil (NFT), a poorly soluble drug, to determine the impact of LLPS on oral absorption behaviors. One of the polymeric excipients used to prepare ASDs, poly(methacrylic acid-co-methyl methacrylate) L100-55, was found to inhibit LLPS. Herein, its impacts on dissolution, transmembrane transport, and oral absorption are discussed to provide a novel methodology to enhance the oral absorption of poorly soluble drugs.

## 2. Materials and Methods

### 2.1. Materials

NFT was purchased from Tokyo Chemical Industry (Tokyo, Japan). Phosphate buffer solution was purchased from FUJIFILM Wako Pure Chemical (Osaka, Japan) and diluted with pure water to obtain a final concentration and pH of 25 mM and pH 6.8, respectively. Vinylpyrrolidone-vinyl acetate copolymer (Kollidon VA64, PVPVA), Eudragit (poly(methacrylic acid-co-methyl methacrylate)) L100-55 (Eudragit), and hydroxypropyl methylcellulose acetate succinate (MG grade, HPMCAS) were supplied by BASF (Ludwigshafen am Rhein, Germany), Evonik (Essen, Germany), and Shin-Etsu Chemical (Tokyo, Japan), respectively. Blank simulated intestinal fluid (SIF) was prepared by dissolving 10 mM sodium chloride and 28.4 mM monobasic sodium phosphate in pure water, which was adjusted to pH 6.5 via the addition of sodium hydroxide solution, where FeSSIF or FaSSIF powder for humans (Biorelevant, London, UK) was dissolved to prepare 3 or 15 mM SIF. All other chemicals were of reagent grade.

### 2.2. Solubility Measurement of NFT

The equilibrium solubility of NFT was determined as a function of pH by adding an excess amount of crystalline NFT to aqueous buffers with different pH values, followed by equilibration at 37 °C for 24 h. The solubilities in 3 or 15 mM SIF were also determined. The effect of the addition of 0.1 *w*/*v*% polymer on solubility was also investigated for each medium. The equilibrated suspensions were filtered using a Millex LG filter (0.20 μm, PTFE). After the first few drops were discarded, the filtrates were collected and diluted with acetonitrile, and centrifugated at 2130× *g* for 10 min to remove the crashed-out polymer. The NFT concentration was determined using a high-performance liquid chromatography (HPLC) system (Shimadzu, Kyoto, Japan), with Unison UK-C18 3 μm, 4.6 × 150 mm (Imtakt, Kyoto, Japan) as a separation column. The mobile phase comprised pH 4.0 phosphate buffer (45%) and methanol (55%), the flow rate was 1.2 mL/min, and the injection volume was 10 μL. The retention time was approximately 8.6 min. The detection wavelength was 283 nm except that 210 nm was applied when the sensitivity of detection was insufficient.

### 2.3. Observation of UV Spectra of NFT/Polymer Mixtures and Determination of Liquid–Liquid Phase Separation (LLPS) Concentration

NFT was dissolved in dimethylsulfoxide (DMSO) at a concentration of 20 mg/mL. Thereafter, the DMSO solution was added to a phosphate buffer (PB) solution with or without 0.1 *w*/*v*% polymer (pH 6.8) at 37 °C with stirring at 300 rpm using the syringe pump, YSP-101 (YMC, Kyoto, Japan), at a feeding rate of 1.2 mL/h. The UV/vis spectra (200–400 nm) with different NFT concentrations and turbidity at 500 nm of solutions were measured on a μDiss ProfilerTM (Pion, Billerica, MA, USA). The concentration at which the turbidity started to increase was defined as the LLPS concentration, which was determined by the intersection of the baseline and the fitting line from the high-concentration side.

### 2.4. Investigation of LLPS Using Polarized Light Microscopy (PLM)

NFT was dissolved in DMSO at a concentration of 20 mg/mL. Approximately 30 μL of the PB solution with or without 0.1 *w*/*v*% polymer was placed on a slide glass, to which 1 μL of the NFT/DMSO solution was added. Then, a cover glass was gently placed on the solution. Observations were carried out at 1 h and 4 h after preparation using an optical microscope (Olympus BX-51, Tokyo, Japan) equipped with a U-POT polarizer and a U-ANT analyzer at 25 °C.

### 2.5. Size and Zeta Potential of the Nanoparticles Formed after LLPS

The size of the particles formed after LLPS was determined using a Zetasizer Nano-ZS (Malvern Instruments, Westborough, MA, USA). The 20 mg/mL solution in DMSO was added to 10 mL of PB solution with or without polymers to achieve NFT concentrations of 10, 30, 60, or 100 μg/mL with stirring at 300 rpm for 3 min at 25 °C. Scattered light from the particles was detected using a backscatter detector at an angle of 173°. The Z-average and zeta potential of the particles were recorded to obtain the average values and standard deviations (*n* = 3). The cumulant method was used for the particle size analysis.

### 2.6. Preparation and Physical Characterization of ASDs

ASDs were prepared using a Büchi B290 mini spray drier (Buchi Labortechnik, Flawil, Switzerland). The feed solution for PVPVA ASD was prepared by dissolving PVPVA and NFT in dichloromethane at concentrations of 10 mg/mL and 50 mg/mL, respectively. The feed solution for HPMCAS ASD was prepared by dissolving HPMCAS and NFT in acetone at concentrations of 10 mg/mL and 40 mg/mL, respectively. For preparing the Eudragit ASD, a solution for spray-drying was prepared by dissolving Eudragit and NFT in acetone/methanol (3/1, *v*/*v*) at concentrations of 5 mg/mL and 20 mg/mL, respectively. Each feed solution was supplied to the nozzle using a peristaltic pump at a rate of 3 g/min. The flow rate of the atomizing air was 350 L/h, which was provided through a 0.7 cm bifluid nozzle. The aspirator was set at 100%, and the inlet temperature was maintained at 100 °C to achieve an outlet temperature of 65 °C. The yield was higher than 70% for all ASDs. The spray-dried powder was subjected to further drying using a vacuum oven (EYELA VOS-310C, Tokyo, Japan) at ca. 0.1 MPa and 40 °C overnight to remove the residual solvent. All ASDs were confirmed to be totally in the amorphous state as presented in the Appendix A, where no diffraction peaks in the X-ray diffraction studies and no melting peaks during the thermal analysis were observed.

### 2.7. Non-Sink Dissolution Study

A 708-DS dissolution apparatus (Agilent, Santa Clara, CA, USA) equipped with paddles was used for the non-sink dissolution study. Crystalline NFT and ASDs were mixed with mannitol (50 *w*/*w*%) to improve their wettability and introduced at a dose of 90 mg (as NFT equivalent) to 900 mL of phosphate buffer (the second fluid of the Japanese Pharmacopeia: JP2, pH 6.8) at 37 °C with stirring at 75 rpm. Approximately 2 mL of the dissolution medium was collected at predetermined time intervals without replenishment with fresh buffer. The NFT concentration was determined in the same manner used for the solubility measurement.

### 2.8. Dissolution and Permeation Assessment Using the Dissolution/Permeation (D/P) System

The dissolution of ASDs and membrane permeation of NFT were evaluated using the D/P system with a Madin–Darby canine kidney (MDCK) II cell monolayer [19]. MDCK II cells were seeded into 6-well plate inserts at a density of 4.5 × 10^5^ cells per insert. The medium was replaced on day 4 and used for the experiments on day 6 after seeding. Further details on the procedure for culturing cells are described elsewhere [19]. The MDCK II cell monolayer in the 6-well plate inserts was equilibrated for 20 min with transport medium (TM, HBSS supplemented with 4.17 mM NaHCO_3_, 10 mM HEPES, and 25 mM glucose adjusted to pH 6.5) for the donor side and bovine serum albumin (BSA) solution (pH 7.4) for the acceptor side. After mounting 6-well plate inserts between the chambers of the D/P system, 8 mL of TM or SIF containing 15 mM sodium taurocholate (TCNa) and 3.75 mM lecithin in TM (15 mM SIF) and 5.5 mL of the BSA solution were added to the donor and the acceptor side, respectively. TCNa and lecithin at these concentrations do not affect the integrity of the MDCK cells [19]. The solutions on both sides of the chamber were stirred at 200 rpm using magnetic stirrers. A physical mixture of crystalline NFT and mannitol (PM), PVPVA ASD, or HPMCAS ASD was introduced to the donor side of the D/P system as suspensions prepared by a homogenizer. D/P chambers were sealed, and the experiments were carried out at 37 °C for 2 h. Solutions were collected from the acceptor side at 30, 60, 90, and 120 min. The donor solutions were also taken at 15 and 120 min and immediately filtered using syringe filters (MultiScreen Solvinertphilic PTFE 0.45μm, Merck, Darmstadt, Germany). The NFT concentration was determined by LC-MS/MS analysis (API4000, SCIEX, UPLC, Waters, Milford, MA, USA). The samples were deproteinized by adding a mixture of acetonitrile and methanol (7/1 (*v*/*v*)), followed by centrifugation at 1400× *g* for 5 min at 25 °C. A Unison UK-C18 ODS (50 mm × 2 mm, 3 μm particle size, Imtakt, Kyoto, Japan) was used as a separation column at 50 °C. The solvents used as mobile phases were (A) 5% acetonitrile including 5 mM ammonium acetate and 0.2% formic acid and (B) 95% acetonitrile including 5 mM ammonium acetate and 0.2% formic acid. The gradient condition was as follows: 0–0.1 min, 0.1% B; 0.1–0.2 min, 0.1–20% B; 0.2–0.4 min, 20–99.9% B; and 0.4–0.9 min, 99.9% B, where the flow rate was 0.8 mL/min. The linearity of the standard curve was confirmed using the concentration range of 0.02–20 μM (r > 0.99).

### 2.9. Oral Administration Study

A protocol for the oral administration study was approved by the Ethical Review Committee of DaiichSankyo RD Novare (Exp. No. 2017-024). The same administration protocol used in our previous studies [15,16] was employed herein, except that the administration dose was 20 mg/10 mL/kg. Briefly, formulations were dispersed in 0.5% methylcellulose (MC) (for crystalline NFT) or purified water (for ASDs), followed by immediate administration to fasted male Crl:CD(SD) rats (6–7-week-old, Charles River Laboratories Japan, Yokohama, Japan, *n* = 3). Approximately 150 μL of the blood samples was withdrawn through the jugular vein at 0.25, 0.5, 1, 2, 4, and 8 h after administration using pre-heparinized syringes. Plasma samples were obtained by centrifugation of the blood at 3000× *g* for 3 min. Thereafter, 20 μL of the sample was mixed with 50% (*v*/*v*) acetonitrile aqueous solution, with 200 μL of acetonitrile/methanol (75/25 (*v*/*v*)) containing 15 ng/mL of niflumic acid added as an internal standard. The NFT concentration was determined using LC-MS/MS, as described for the D/P study.

## 3. Results

### 3.1. Solubility and Phase Separation Study

Figure 1 shows the solubility of NFT in each medium. The solubility, *S*, in buffered solutions was fitted with the modified Henderson–Hasselbalch equation:(1)$$S=S_{0}\left(1+\frac{\left\lceil H^{+}\right\rceil }{K_{a}}\right)$$where *S*_0_ is the equilibrium solubility of the undissociated species in the aqueous media and *K_a_* is the acidity constant. A p*K*a value of 7.3 [20] was used for fitting. The solubilities in the pH range of 3 to 7 followed Equation (1), which indicated NFT was dissolved in a monomeric state in this pH range. However, the solubility at pH 1.2 was excluded from the fitting as it was extraordinarily lower than expected; this was most likely to be due to the formation of the hydrochloride salt [20]. Table 1 summarizes the solubilities in various media. The increase in solubility in SIF might be due to solubilization by bile salt micelles. HPMCAS and PVPVA did not affect solubility; however, solubility slightly increased with the addition of Eudragit. Figure 2 shows an increase in turbidity with the increasing NFT concentration, where the LLPS concentrations are indicated by arrows. In buffered solutions, LLPS occurred at 45 μg/mL and remained almost the same in the presence of PVPVA or HPMCAS. However, the turbidity gradually increased without showing a breakpoint in the presence of Eudragit.

### 3.2. PLM Observation of LLPS Behavior

Figure 3 shows the LLPS behavior under an optical microscope. Note that the phase separation behavior in this study (on the glass slides) was slower than that in bulk, which might be due to the effect of the confined atmosphere [21]. In addition, something that should be stressed is that the NFT concentration in this study was higher than that for the particle size analysis shown next by more than an order of magnitude for the easy detection of the LLPS droplets and crystals. The spherical dispersed phase was investigated initially, regardless of the absence or presence of polymers, and was found to be maintained for at least for one hour. No crystalline structures were observed during this period. The size of the droplets was smaller in the presence of polymers than in polymer-free media. After four hours, large crystals were observed in the absence of polymers, whereas small crystals appeared in the PVPVA and HPMCAS solutions. However, nearly spherical but irregular shapes of aggregated structures were found for the Eudragit solution without the presence of crystalline particles. Thus, only Eudragit could suppress the crystallization of NFT after supersaturation.

### 3.3. Particle Size and Zeta-Potential of Particles Observed in Supersaturated Solutions

The particle size, polydispersity index (PDI), and zeta potential as a function of NFT concentration with or without a polymer are shown in Table 2. The particle size ranged from 200 to 400 nm when the NFT concentration was below 60 μg/mL for all of the solutions. The diameter increased with the increasing NFT concentration for the polymer-free and PVPVA solutions. Charged polymers (HPMCAS and Eudragit) suppressed the increase in particle size, which might be due to electrostatic repulsion between the particles. The zeta potentials of the particles in the absence of polymers and in the presence of PVPVA were almost neutral, whereas those in the presence of HPMCAS and Eudragit exhibited negative values. The electrostatic repulsion seemed to be stronger in the presence of Eudragit compared to HPMCAS, similar to the cases for LLPS particles of other drugs [14,15,16].

### 3.4. Peak Shift of UV Spectrum

To investigate the interaction between NFT and polymers, the concentration-dependent UV spectrum of NFT was observed under a constant polymer concentration. The maximum absorption was found at 280 nm for buffered, HPMCAS, and PVPVA solutions, regardless of the NFT concentration. This was also the case for the Eudragit solution when the NFT concentration was below 20 μg/mL; however, the absorption maxima were red-shifted above that NFT concentration (Figure 4). Thus, the polarity around NFT molecules might increase with the addition of Eudragit, indicating strong molecular interactions between NFT and Eudragit.

### 3.5. Preparation and Characterization of ASDs

The XRPD and DSC data of the ASDs are provided in the Appendix A. The onset glass transition temperatures (*T*g) for PVPVA, HPMCAS, and the Eudragit ASDs were 82, 85, and 118 °C, respectively. Only one *T*g was observed for each ASD. These *T*gs were markedly higher than those of the neat NFT glass (25 °C) and between those of NFT and polymers, suggesting molecular mixing of NFT and the polymers.

In the non-sink dissolution study, all ASDs exhibited better dissolution than neat NFT (Figure 5). PVPVA and the HPMCAS ASD rapidly dissolved to reach almost the phase separation concentration within 5 min. The slight difference between LLPS and the plateau concentrations is explained by the shift in the equilibrium state owing to the presence of excess solids [15]. In contrast, the observed concentration of NFT achieved by the Eudragit ASD was only three-fold higher than that of the neat NFT crystal, which was significantly lower than the LLPS concentration of NFT. The dissolved NFT concentrations were stable for all of the ASDs, indicating that crystallization of the NFT was not likely to occur during the investigation.

### 3.6. D/P System

PM and ASDs prepared with two different acidic polymers (HPMCAS and Eudragit) were subjected to the D/P study. Crystalline NFT and HPMCAS ASD dissolved immediately to reach a metastable concentration within 15 min, which was maintained for 120 min in both TM and 15 mM SIF. The Eudragit ASD dissolved relatively slowly compared to PM and HPMCAS ASD; therefore, the concentrations at 120 min were slightly higher than those at 15 min. The detailed concentration data are provided in the Appendix A. Figure 6a shows the NFT concentration in the donor chamber after 120 min as a function of the applied NFT dose in the TM. The NFT concentration increased with an increase in the dose of HPMCAS ASD. At the highest dose (3.6 mg), LLPS was almost attained in the presence of HPMCAS despite the slightly lower concentration than LLPS; this is due to the change in equilibrium balance, as observed in the non-sink dissolution study. The dissolved NFT concentration from the Eudragit ASD in the dissolution chamber peaked at a dose of 0.4 mg. The concentration decreased with increasing doses above this dose. The flux had an almost linear relationship with the dissolved concentration, as shown in Figure 6b. In 15 mM SIF, the trend between the added dose and the dissolved concentration was the same as that in TM, except that the dissolved concentration was much higher than that in TM (Figure 6c). The higher NFT concentration than the LLPS concentration suggested solubilization of NFT by the SIF components. The flux in 15 mM SIF also exhibited a correlation with the dissolved concentration below 50 μg/mL (Figure 6d); however, the flux was significantly smaller relative to that with TM (Figure 6b).

### 3.7. Oral Absorption Study

Figure 7 shows the plasma concentration profile following the oral administration of NFT from the PM and ASDs. The pharmacokinetic parameters are summarized in Table 3. PVPVA and HPMCAS ASDs did not improve the oral absorption of NFT compared with PM. However, its oral absorption was improved by using the Eudragit ASD, with slightly higher Cmax and AUC levels than those obtained for other formulations. Such a finding is despite a much lower dissolved concentration of NFT when the Eudragit ASD was employed relative to other ASDs (Figure 5). XRPD measurements of the remaining formulation at 2 h after oral administration confirmed crystallization only for the PVPVA ASD.

## 4. Discussion

### 4.1. LLPS and Its Inhibition in the Presence of Polymers

Generally, hydrophilic polymers do not change or only slightly increase the solubility of poorly soluble drugs, as observed for PVPVA and HPMCAS (Table 1). However, the solubility of NFT increased in the presence of Eudragit relative to the other two polymers, indicating a strong interaction between them. LLPS is basically not influenced by the presence of polymers [14], as observed for PVPVA and HPMCAS. However, LLPS was not clearly observed in the presence of Eudragit. The red-shift of the UV spectra of NFT in the presence of Eudragit (Figure 4) suggested that it offered a hydrophilic environment for NFT, presumably because of its charged moiety, that is, NFT and Eudragit were the most likely to be associated. Such an assumption was supported by the gradual increase in turbidity without a break point with the increasing concentration of NFT. Thus, Eudragit might inhibit LLPS owing to its strong molecular interaction with NFT. Note that the pH change by the addition of acidic polymers such as Eudragit was not likely to be the reason for shift of the UV spectra, as the pH change of the phosphate buffers by adding each polymer was within 0.1. In the particle size measurement (Table 2), the Eudragit/NFT particles had almost the same size as the HPMCAS/NFT LLPS particles. However, the particle formation mechanism might differ between the two cases. The particles in the HPMCAS solution are likely to be formed from the LLPS droplets, whereas those in the Eudragit solution should be originate from molecular aggregates with NFT.

### 4.2. Non-Sink Dissolution Behaviors of ASDs

The dissolution study was conducted under a non-sink condition, where the applied NFT dose was larger than the LLPS concentration by more than two-fold and 50-to-80-fold the equilibrium crystalline solubility. The dissolved NFT concentration from PVPVA and HPMCAS ASDs immediately reached almost the same level as the LLPS concentration, whereas NFT appeared to be released slowly from the Eudragit ASD. This was not likely to be because of the slow disintegration of the formulation as the Eudragit ASD could be immediately dispersed in the media. Moreover, crystallization of the drug, which is generally one of the reasons for the poor dissolution property of ASD, was not observed during the dissolution study. The apparent slow release might be explained by the formation of the molecular complex of NFT and Eudragit, which may be removed during filtration using syringe filters if the size of the molecular complex is larger than the pore size of the filter. In fact, DLS studies revealed the formation of aggregates, of which the size depended on the concentration and was comparable to the pore size of the syringe filters. Similar observations can be found in the literature, as exemplified by the observation that the free fraction of dissolved ibuprofen was found to decrease upon complexation with Eudragit [22]. The NFT molecules associated with Eudragit may or may not pass through the membrane, which may be influenced by the strength of the association.

### 4.3. Membrane Permeation Behaviors from ASDs

In the D/P studies, the donor concentration of NFT was evaluated in the same manner as the concentration used for the dissolution study. Thus, the molecular aggregates formed in the presence of Eudragit might be removed during the evaluation. Both HPMCAS and Eudragit ASDs showed improved solubility and increased flux through the cell membrane compared with those of NFT crystals at a dose of 3.6 mg. The improvement in dissolution and flux through the membrane by the HPMCAS ASD was more remarkable than the results of the Eudragit ASD, which can be explained by the complexation of NFT with Eudragit.

In 15 mM SIF, the dissolved NFT concentrations were much higher in the presence of both polymers than those in bile-free TM. Thus, NFT seemed to be preferentially distributed to bile micelles rather than the formation of the molecular complex with Eudragit. The dissolved concentration in the presence of Eudragit aligned with that of HPMCAS when the applied dose was lower than 1.2 mg (Figure 6c). However, the concentration decreased dramatically when the applied NFT increased to 3.6 mg only in the Eudragit system, suggesting a competitive role for the molecular complexation. Both solubilization by bile micelles and complexation with Eudragit seemed to inhibit membrane permeation as the amount permeated in the SIF was much lower than that in the bile-free TM. The free fraction can be estimated by subtracting the solubility in bile-free media from that with bile micelles. Figure 8 shows the relationship between the calculated free fraction and the observed flux in both TM and SIF, which exhibits a good correlation. Thus, only the free fraction might pass through the cell membrane.

Biorelevant media have recently been used as the gold standard in the purpose of predicting the oral absorption of drugs from dissolution tests. However, the dissolution properties observed in such media frequently fail to provide accurate predictions, which is partially due to the variable contribution of the intestinal colloids to incorporate drugs [23]. Our result appears to contradict the results of Lu et al., whereby the transmembrane flux of telaprevir was not influenced by the addition of TCNa [17]. On the other hand, a similar experiment was carried out using octreotide by Dening et al. who found that the flux decreased by ca. 43% with the addition of 15 mM TCNa; however, the effect was weaker than that of dihydroxy bile salts, such as sodium taurodeoxycholate [24]. As the interaction strength between drugs and bile salt micelles depends on the drug species [25], the inhibitory effect on membrane permeation may also vary [23], which is an aspect on which further systematic investigations are required.

### 4.4. Oral Absorption Study

Both dissolution and D/P studies indicated that HPMCAS was a better choice than Eudragit as an excipient for NFT ASD to improve its oral absorption. However, the increase in the absorption of NFT was only observed for the Eudragit ASD, although there is an apparent poor dissolution behavior and low flux of NFT through the cell membrane from this ASD. Thus, the molecular complex composed of NFT and Eudragit seems to be easily disintegrated to release NFT for absorption. Another assumption on the effect of Eudragit may be the inhibition of crystallization of NFT. Although all polymers seemed to have an inhibition effect of crystallization based on the dissolution and D/P studies, the crystallization behavior of NFT may be different in the gastrointestinal tract, i.e., crystallization may be effectively suppressed in the presence of Eudragit by forming a molecular complex. In fact, the PLM study suggested the strong inhibitory effect of Eudragit for the crystallization of NFT.

When absorption from ASD is limited by solubility, the absorption is determined by the LLPS concentration [14]. As the LLPS is generally higher than the equilibrium solubility by more than ten-fold, the absorption can be greatly improved by using ASDs. On the other hand, LLPS is regarded as the upper limit of solubility achievable by amorphization. In the case of NFT, approximately 40 μg/mL is the maximum concentration achievable in the gastrointestinal tract, if ASDs that can produce LLPS particles are used. The complexation of drugs and excipients is generally recognized as an unfavorable phenomenon, because it decreases transmembrane flux [18]. However, we demonstrated that it can be advantageous for oral absorption. The NFT concentration can be much higher than 40 μg/mL, if the LLPS is inhibited. As observed in the D/P system study using 15 mM SIF, the complex between NFT and Eudragit might be disintegrated in the presence of bile micelles in the small intestine. Thus, the formation of a molecular complex with an excipient polymer could be a strategy to break the solubility ceiling (LLPS) to improve the oral absorption of poorly soluble drugs.

## 5. Conclusions

In this study, dissolution, membrane permeation, and oral absorption behavior were observed for NFT ASDs prepared using PVPVA, HPMCAS, and Eudragit as excipient polymers. In the presence of PVPVA and HPMCAS, the LLPS concentration of NFT was not influenced. The ASDs prepared using these polymers did not enhance the oral absorption of NFT. In contrast, LLPS was not observed in the presence of Eudragit. Only ASD prepared by using this polymer improved the oral absorption of NFT. Eudragit might form a molecular complex with NFT, which might be disintegrated in the presence of bile micelles in the SIF or small intestine. Altogether, inhibition of LLPS by molecular complex formation could be a powerful strategy to improve the oral absorption of poorly soluble drugs.

## Figures and Tables

**Figure 1 pharmaceutics-14-02664-f001:**
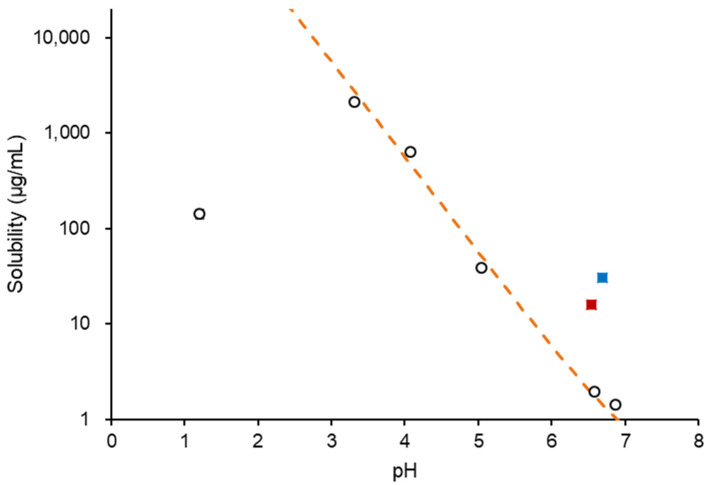
pH-solubility profile of NFT. (○) Buffered solutions, (■) 3 mM SIF, and (■) 15 mM SIF.

**Figure 2 pharmaceutics-14-02664-f002:**
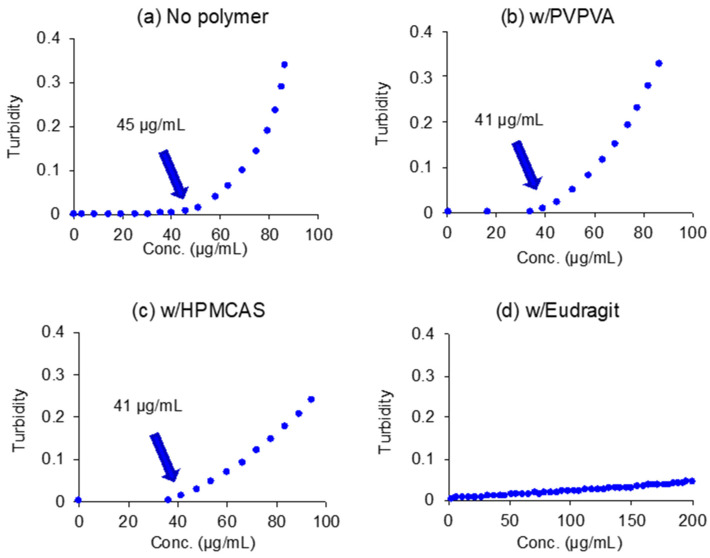
Turbidity of the NFT solutions/suspensions. LLPS concentrations are indicated in the figures.

**Figure 3 pharmaceutics-14-02664-f003:**
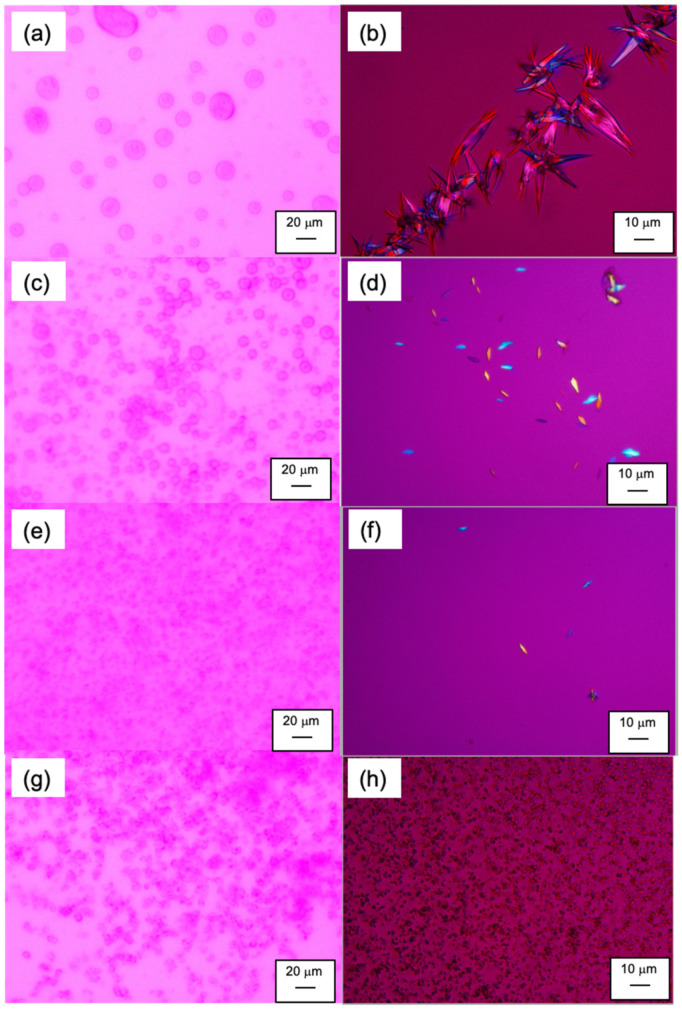
PLM images of LLPS and crystallization from supersaturated NFT solutions: (**a**) *w*/*o* polymer, 1 h after mixing; (**b**) *w*/*o* polymer, 4 h after mixing; (**c**) *w*/0.1% PVPVA, 1 h after mixing; (**d**) *w*/0.1% PVPVA, 4 h after mixing; (**e**) *w*/0.1% HPMCAS, 1 h after mixing; (**f**) *w*/0.1% HPMCAS, 4 h after mixing; (**g**) *w*/0.1% Eudragit, 1 h after mixing; (**h**) *w*/Eudragit, 4 h after mixing.

**Figure 4 pharmaceutics-14-02664-f004:**
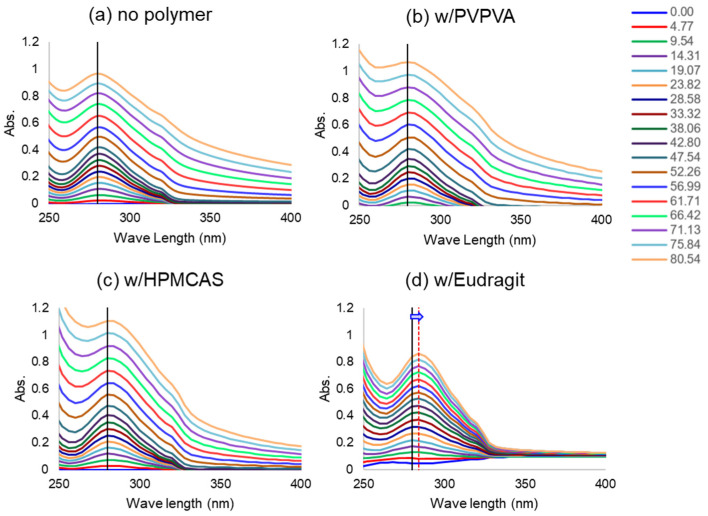
Effect of coexisting polymers on the UV spectra of NFT. The NFT concentrations (μg/mL) are shown in the figure (the concentration increased from bottom to top). The absorption maximum and its shift in the presence of Eudragit are presented by vertical lines in the figure.

**Figure 5 pharmaceutics-14-02664-f005:**
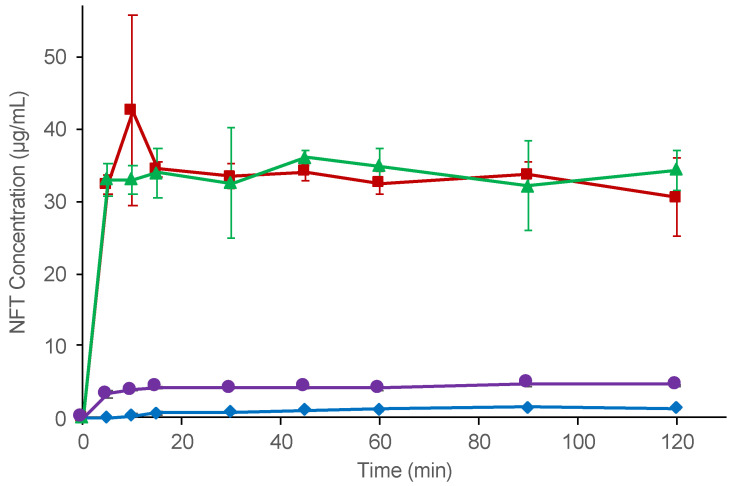
Non-sink dissolution study of crystalline NFT and NFT ASDs. Symbols: crystalline NFT (◆), PVPVA ASD (■), HPMCAS ASD (▲), and Eudragit ASD (●).

**Figure 6 pharmaceutics-14-02664-f006:**
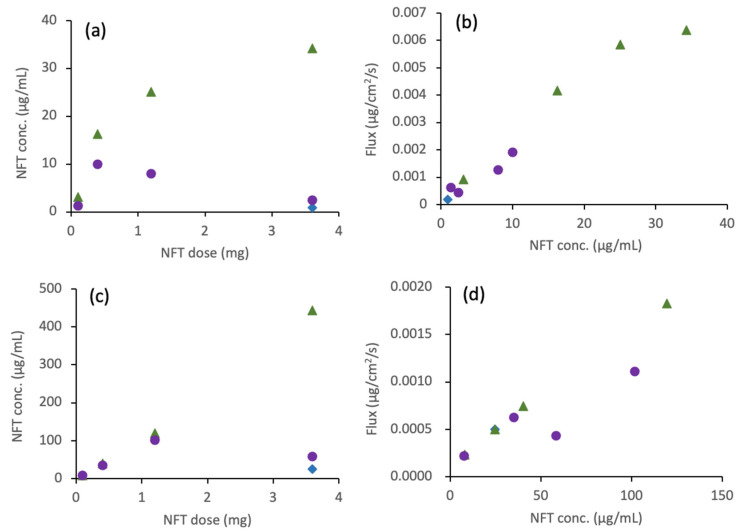
D/P study of NFT ASDs: (**a**) dissolved NFT concentration in a donor chamber after 120 min as a function of the applied NFT dose. TM was used as a medium. (**b**) Transmembrane flux as a function of NFT concentration in the donor phase. TM was used as a medium. (**c**) Same as (**a**) except that SIF was used as a medium. (**d**) Same as (**b**) except that SIF was used as a medium. Symbols: crystalline NFT (◆), HPMCAS ASD (▲), and Eudragit ASD (●).

**Figure 7 pharmaceutics-14-02664-f007:**
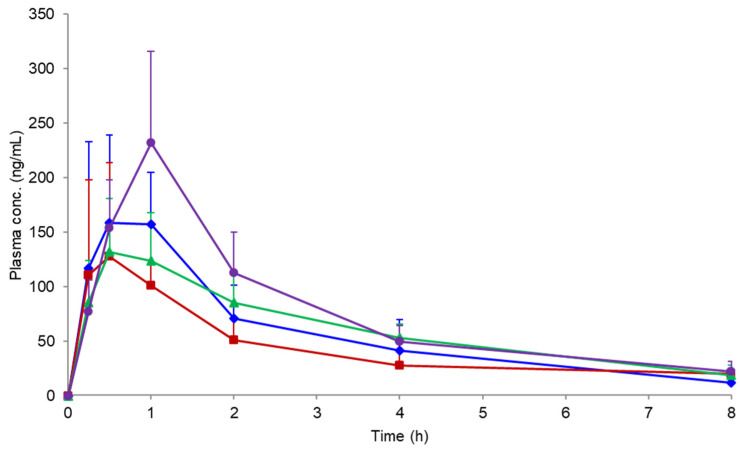
Plasma NFT concentration profiles after the oral administration of crystalline NFT and its ASDs in rats at a dose of 20 mg/kg (*n* = 3). Symbols: crystalline NFT (◆), PVPVA ASD (■), HPMCAS ASD (▲), and Eudragit ASD (●).

**Figure 8 pharmaceutics-14-02664-f008:**
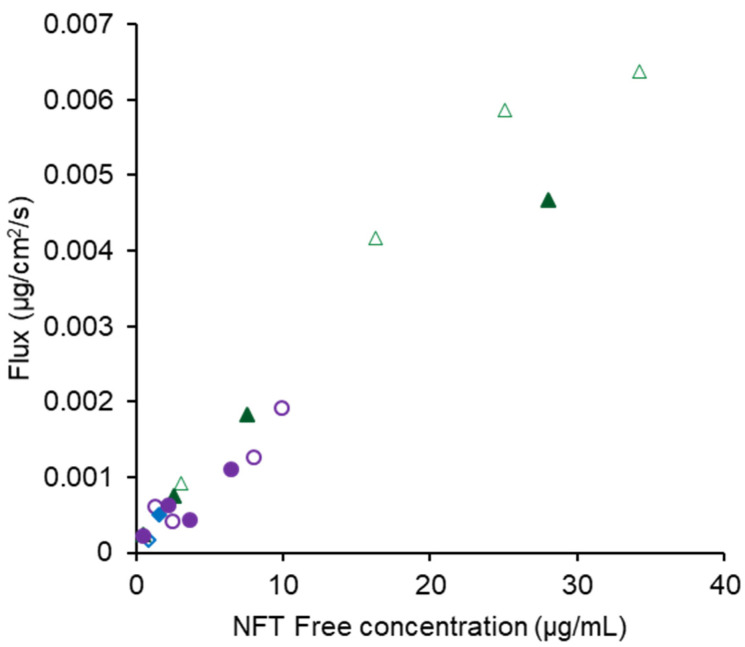
Correlation between free NFT concentration and transmembrane flux. The symbols are the same as those in Figure 6, except that open and close symbols represent the data obtained for TM and SIF as media.

**Table 1 pharmaceutics-14-02664-t001:** Solubility of NFT in polymer solutions (μg/mL).

Medium	pH	Solubility *w*/*o* Polymer	*w*/0.1 *w*/*v*% PVPVA	*w*/0.1 *w*/*v*% HPMCAS	*w*/0.1 *w*/*v*% Eudragit
Phosphate Buffer	6.8	1.44 ± 0.00	1.59 ± 0.02	1.35 ± 0.05	1.95 ± 0.06
TM	6.5	1.54 ± 0.10	NT *	1.84 ± 0.06	2.76 ± 0.10
15 mM SIF	6.5	30.8 ± 0.1	NT *	30.2 ± 0.4	35.0 ± 0.2

* NT: not tested.

**Table 2 pharmaceutics-14-02664-t002:** Particle size and zeta potential of NFT/polymer nanostructures.

Polymer	NFT Concentration (μg/mL)	Mean Diameter (nm)	PDI	Zeta Potential (mV)
None	60	400 ± 8	0.264 ± 0.076	−3.1 ± 1.3
	100	607 ± 43	0.065 ± 0.055	+8.1 ± 2.3
0.1 *w*/*v*% PVPVA	60	297 ± 9	0.246 ± 0.003	−0.1 ± 0.4
	100	401 ± 7	0.130 ± 0.036	+0.2 ± 0.3
0.1 *w*/*v*% HPMCAS	60	271 ± 4	0.160 ± 0.044	−12.6 ± 1.4
	100	277 ± 12	0.080 ± 0.001	−12.8 ± 0.6
Eudragit	10	162 ± 12	0.544 ± 0.024	−30.9 ± 0.5
	30	211 ± 10	0.321 ± 0.006	−29.4 ± 1.2
	60	215 ± 2	0.267 ± 0.016	−29.0 ± 2.1
	100	267 ± 19	0.228 ± 0.027	−27.2 ± 2.6

**Table 3 pharmaceutics-14-02664-t003:** Pharmacokinetic parameters after the oral administration of NFT formulations.

	Tmax ^a^ (h)	Cmax ^b^ (ng/mL)	AUC0-8h ^b^ (ng·h/mL)	AUC0-∞ ^b^ (ng·h/mL)	t1/2 ^b^ (h)
PM	0.50	177 ± 75	438 ± 153	483 ± 142	2.6 ± 0.8
PVPVA ASD	0.50	130 ± 83	345 ± 176	664 ± 354	9.1 ± 10.0
HPMAS ASD	0.50	132 ± 49	475 ± 152	555 ± 200	2.7 ± 0.3
Eudragit ASD	1.0	232 ± 84	590 ± 163	685 ± 168	2.8 ± 0.8

^a^ median, ^b^ mean ± SD.

## Data Availability

The data presented in this study are available upon request.

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
