# Peer review of "Inhibition of Liquid–Liquid Phase Separation for Breaking the Solubility Barrier of Amorphous Solid Dispersions to Improve Oral Absorption of Naftopidil"

_pharmaceutics, 2022, doi:10.3390/pharmaceutics14122664_

Round 1
Reviewer 1 Report
This is an interesting study, invesitgating the effect of LLPS not only on drug solubility but also how this phenomenon may affect drug permeation. In principle, this paper is suitable to be published in Pharmaceutics but I recommend some minor revision to strengthen the manuscript and to provide clarity on some aspects which are not clear and open for interpretation in the current format.
Lines 40 - 41: "Solubility and the dissolution rate are the...." please review this sentence and correct, the structure is not correct.
Lines 73-72: What does Ca Sub signify?
Lines 106-107: It is unclear where the crashed-out polymer fits in with the experimental setup? The paragraph states that it is the experimental setup to determine the equilibrium solubility of NFT at various pH levels. It is mentioned that an excess amount of NFT was added to each aqueous buffer, after a period of time the solutions were filtered. The filtrates were then diluted with ACN and then centrifuged. How did the authors ensure that only the polymer crashed out? Does the possibility exist that the NFT could've also crashed out with the polymer?
Lines 111-112: It is mentioned that the detection wavelength was either 210 or 283. May the authors please specify why two wavelengths were utilised?
Paragraph 2.4 - The authors should justify why DMSO was used to investigate the LLPS in DMSO. Why is this solvent different from the other solvents used throughout this study. Is it possible to correlated these results obtained from DMSO with results obtained from aqueous buffered solutions? How does this correlated with the quantified LLPS described throughout Figure 2.
Line 154: Mention is made that the spray dried samples were subjected to vacuum at 40°C. It is assumed that this was done utilising a vacuum oven? The authors must specify the type of oven and relevant settings.
Line 156: It was mentioned that all prepared ASDs were confirmed to be in the amorphous state. How was this confirmed?
Line 161: The drug or ASDs were mixed with mannitol. What was the rationale for doing this?
What is the rationale for using MDCK II cells during the dissolution and permeation assessment?
It is unclear why the authors opted to calculate solubility of NFT using the Henderson-Hasselbalch equation. This must be clarified.
Author Response
Authors’ response to Reviewer 1
We have appreciated reviewers’ comment, which helped a lot for improving our manuscript. Please find reply comments below.
Lines 40 - 41: "Solubility and the dissolution rate are the...." please review this sentence and correct, the structure is not correct.
We have revised this sentence (line 40- 41).
Lines 73-72: What does Ca Sub signify?
We apologize for insufficient information. We have spelled out fill name for this compound (line 83-84).
Lines 106-107: It is unclear where the crashed-out polymer fits in with the experimental setup? The paragraph states that it is the experimental setup to determine the equilibrium solubility of NFT at various pH levels. It is mentioned that an excess amount of NFT was added to each aqueous buffer, after a period of time the solutions were filtered. The filtrates were then diluted with ACN and then centrifuged. How did the authors ensure that only the polymer crashed out? Does the possibility exist that the NFT could've also crashed out with the polymer?
This is a very important point. We totally agree that there is a possibility that the drug may be trapped in the precipitated polymer during the dilution process using organic solvents. However, we recognize that it does not happen in this case. We are not sure if the polymer concentration is much higher, but at a concentration of 0.1% of polymer, we recognize that almost all the drug is fully recovered.
Lines 111-112: It is mentioned that the detection wavelength was either 210 or 283. May the authors please specify why two wavelengths were utilised?
The analysis at the wavelength of 210 nm is more sensitive compared with that at 283 nm. However, this wavelength is not specific for NFT. Therefore, only when the detection at 283 nm was difficult, the wavelength of 210 nm was applied. This point was clarified in the text. (line 125-126)
Paragraph 2.4 - The authors should justify why DMSO was used to investigate the LLPS in DMSO. Why is this solvent different from the other solvents used throughout this study. Is it possible to correlated these results obtained from DMSO with results obtained from aqueous buffered solutions? How does this correlated with the quantified LLPS described throughout Figure 2
We apologize for insufficient explanation. We have investigated LLPS in buffered solutions. DMSO was used only for adding NFT to the buffered solutions. We have revised explanations to clarify this point (line 131, 141 and 150).
Line 154: Mention is made that the spray dried samples were subjected to vacuum at 40°C. It is assumed that this was done utilising a vacuum oven? The authors must specify the type of oven and relevant settings.
We have added information on the vacuum oven used and clarified relevant settings in line 172-173.
Line 156: It was mentioned that all prepared ASDs were confirmed to be in the amorphous state. How was this confirmed?
We have clarified that how we confirmed the ASDs were in an amorphous state in line 173-176.
Line 161: The drug or ASDs were mixed with mannitol. What was the rationale for doing this?
It was to improve wettability of neat drug and ASDs. Without this, the ASDs may form large agglomerations. We have added the explanation in the text (line 181).
What is the rationale for using MDCK II cells during the dissolution and permeation assessment?
This is an already established protocol to predict in vivo absorption as reported in reference 19.
It is unclear why the authors opted to calculate solubility of NFT using the Henderson-Hasselbalch equation. This must be clarified.
Henderson-Hasselbalch equation explains pH-dependent solubility very well, if the compound is ideally dissolved in a monomeric state. Therefore, this attempt was made to confirm dissolved state of NFT. Additional explanation was made in the text to clarify this point (line 248-249).
Reviewer 2 Report
Recommendation: Publish after minor revisions noted.
Comments:
Authors systematically explored the dissolution, membrane permeation, and oral absorption behavior of NFT ASDs prepared using PVPVA, HPMCAS, and Eudragit as excipient polymers. The paper is well written, and I would like to recommend it to be published with some revision. Please see my comments below.
1. Please explain why the particle size of LLPS in Figure 3 micro size while it is nano size in Table 2.
2. Is it possible to characterize the the complex between NFT and Eudragit by some techniques,such as SEM or TEM?
3. Particle size of complex between NFT and Eudragit is nano size showed in Table 2. Is it possible that complex and LLPS occur at the same time?
4.The method of UV spectrum did not provide.
5. Please explain why the UV spectrum can analyse molecular interactions, the pH change of solution may also influence the UV spectrum and the addition of polymer may influence the pH.
6. Eudragit inhibited the crystallization rate, it may be the reason that it can improve the oral absorption.
Author Response
Authors’ response to Reviewer 2
We have appreciated reviewers’ comment, which helped a lot for improving our manuscript. Please find reply comments below.
- Please explain why the particle size of LLPS in Figure 3 micro size while it is nano size in Table 2.
It was because of difference in the NFT concentration. In the microscopic observation, high concentration was required for the detection of LLPS droplets and particles. We appreciate reviewer’s comment, as this information should be stressed in the manuscript. Additional explanation was made in the text (line 274-277).
- Is it possible to characterize the complex between NFT and Eudragit by some techniques, such as SEM or TEM?
Although we also think that further characterization of the molecular complex is necessary, it has not been successful so far. As the morphology can change if the dispersions are dried, SEM is difficult to use. Use of cryo-TEM may have some chance; however, characterization of the molecular complex should be difficult. Although we expected much to NMR, it was also difficult presumably because the complex had solid-like property. We would like to leave this subject for further study.
- Particle size of complex between NFT and Eudragit is nano size showed in Table 2. Is it possible that complex and LLPS occur at the same time?
This is very interesting guess, but in this case, only molecular complex should exist for the NFT/Eudragit system based on the turbidity study. This is why we use the ambiguous word “nanostructures” in the title of Table 2. The observed particle size for the NFT/Eudragit system should be that of the molecular complex.
4.The method of UV spectrum did not provide.
It was provided, but we thought that the comment by the reviewer was natural, as it was only by one sentence and very difficult to find. We have revised the section title to solve this problem. (line 128)
- Please explain why the UV spectrum can analyse molecular interactions, the pH change of solution may also influence the UV spectrum and the addition of polymer may influence the pH.
UV spectrum exhibits red- and blue-shift under hydrophilic and hydrophobic atmospheres, respectively (line 394-396). Therefore, it can provide information of distance of molecules. We agree that the pH change of solution could impact UV spectrum. However, we confirmed that the pH change of the phosphate buffer by adding 0.1 w/v% of polymers were within 0.1. We have added explanation on this point (line 403-406).
- Eudragit inhibited the crystallization rate, it may be the reason that it can improve the oral absorption.
All the polymers used in this study exhibited inhibitory effect of crystallization of NFT. However, as the crystallization behavior of NFT may be different in the gastrointestinal tract, the assumption by the reviewer may have a chance. We have added this to the text (line 477-481).